# cHeartFlow: Synthesizing cardiac MR images from sketches

**Xinrui Zu**[1]                                                    ZUXINRUI95@GMAIL.COM

**Qian Tao**[1]                                                    Q.TAO@TUDELFT.NL

[1] *Department of Imaging Physics, Delft Unversity of Technology*

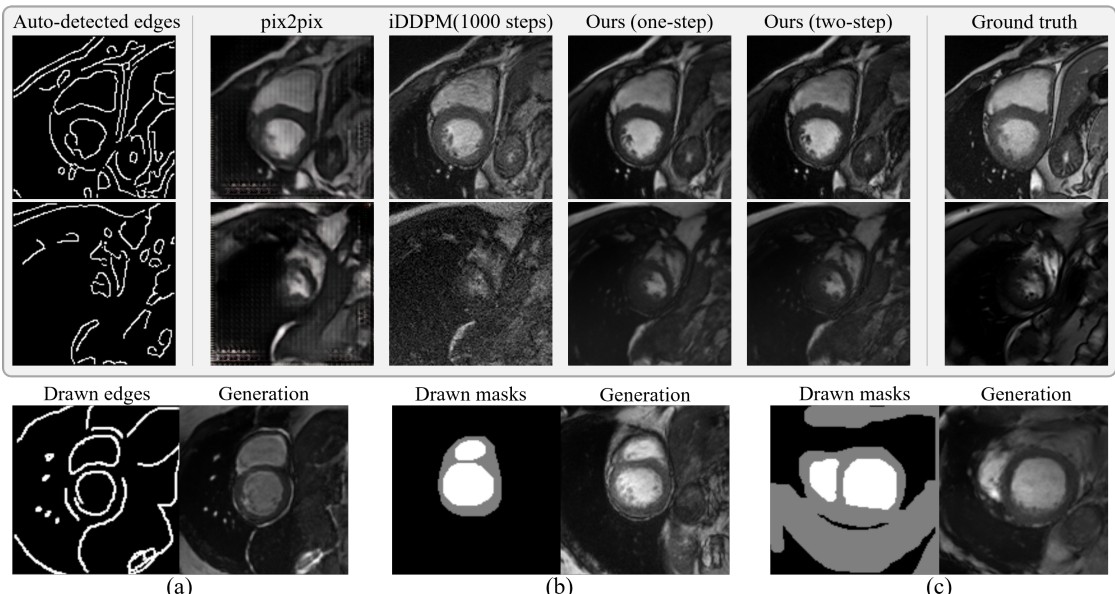

Figure 1: Upper: CMR images synthesized by cHeartFlow, in comparison with the baselines. The models are trained on the ACDC dataset and tested on the M&Ms dataset to demonstrate generalizability. Lower: cHeartFlow-synthesized image with hand-drawn sketches. (a) is trained with auto-detected Canny edges. (b) and (c) are trained with auto-generated segmentations using Segment Anything model (SAM). cHeartFlow accepts diverse input in the form of simple sketches (edges, masks).

## Abstract

Medical image synthesis is a highly promising approach to generate and augment medical data, which suffers from high acquisition costs and stringent privacy restrictions. However, current generation methods typically require detailed anatomical annotations and are limited in generating high-quality, anatomy-compliant images. To overcome the limitations, we present contrastive HeartFlow (cHeartFlow), a novel generative framework to synthesize cardiac magnetic resonance (CMR) images from simple sketches by training on contrastive pairs of images and sketches. cHeartFlow supports one-step synthesis and allows multi-step synthesis for a flexible trade-off between faithfulness and realism. We illustrate the effectiveness and generalizability of cHeartFlow through our experiments on different input sketches, compared with GAN-based and diffusion-based baselines.

**Keywords:** Medical image synthesis, contrastive learning, cardiac MRI

# 1. Introduction

With the recent breakthrough of diffusion models in multi-modal data generation (Kaze-rouni et al., 2023), synthesizing medical images by generative models has emerged as a promising alternative for curating and augmenting medical image data. Firstly, the paucity of medical images due to high acquisition costs and privacy concerns creates a major bottle-neck in machine learning research that depends on the size and diversity of data. Secondly, producing an unlimited resource of synthetic medical images has shown educational value for teaching and training radiologists (Skandarani et al., 2023). Unfortunately, current meth-ods suffer from insufficient image generation quality, while requiring extensive anatomical annotations as input for supervised generation (Singh and Raza, 2021; Jiang et al., 2023).

In this paper, we introduce contrastive HeartFlow (cHeartFlow), which can convert coarse image sketches into clinically realistic CMR images, based on a consistency-model-based method (Song et al., 2023) with a novel contrastive learning and self-augmentation sampling strategy. cHeartFlow trades off between anatomical faithfulness and image re-alism. We evaluated the effectiveness and generalizability of cHeartFlow with different coarse sketches (edges/segmentations) as input, which can be automatically generated, and out-of-distribution hand-drawn sketches.

# 2. Method

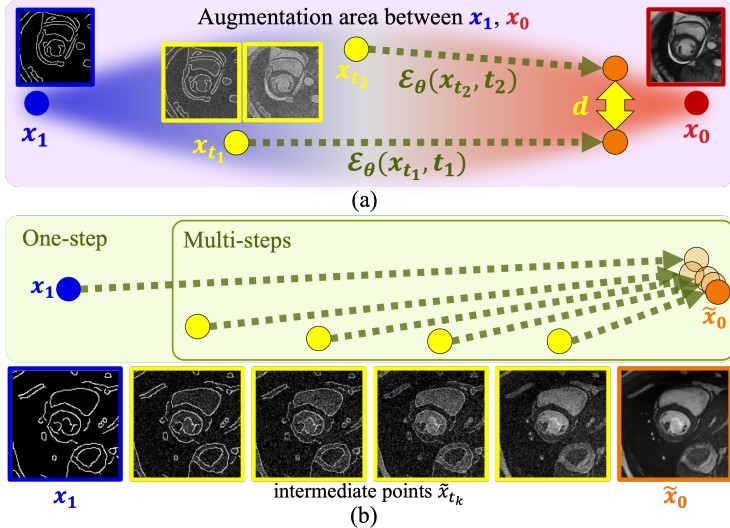

(a)

One-step   Multi-steps

$x_1$

$\tilde{x}_0$

$x_1$   intermediate points $\tilde{x}_{t_k}$   $\tilde{x}_0$

(b)

Figure 2: An overview of the proposed method in training and sampling. (a) The train-ing minimizes the distances between the encodings of the contrastive pairs, which are sampled in the augmentation area between a CMR $\mathbf{x}_0$ and its auto-generated sketches $\mathbf{x}_1$ (Eq.2). (b) Self-augmentation sampling allows one-step gen-eration to obtain $\tilde{\mathbf{x}}_0^{(1)}$, or op-tionally multi-step generation to obtain $\tilde{\mathbf{x}}_0^{(k)}$ (Eq.3/4).

cHeartFlow is built upon contrastive learning and consistency models, with both meth-ods minimizing a distance function $d(\cdot, \cdot)$. The loss function is formulated as:

$$\mathcal{L}_{\text{cHeartFlow}}(\mathcal{E}_\theta) = d\big(\mathcal{E}_\theta(\mathbf{x}_{t_1}, t_1), \mathcal{E}_\theta(\mathbf{x}_{t_2}, t_2)\big), \qquad 0 \le t_1 < t_2 \le 1 \tag{1}$$

where $\mathcal{E}_\theta$ is the generative encoder to be trained. $\mathbf{x}_{t_1}$ and $\mathbf{x}_{t_2}$ are the contrastive pair augmented by the noisy interpolation between a CMR $\mathbf{x}_0$ and its auto-generated coarse sketch $\mathbf{x}_1$:

$$\mathbf{x}_t = (1 - t)\mathbf{x}_0 + t\mathbf{x}_1 + t(1 - t)\sigma^2\epsilon, \qquad 0 < t < 1 \tag{2}$$

where $\epsilon \sim \mathcal{N}(\mathbf{0}, \mathbf{I})$ denotes the intermediate noise of the augmentation area between $\mathbf{x}_0$ and $\mathbf{x}_1$ with diffusion scale $\sigma$. With a well-trained model under Eq.1, we can generate a CMR image from coarse sketches through one-step sampling $\tilde{\mathbf{x}}_0 = \mathcal{E}_\theta(\mathbf{x}_1, 0)$. Alternatively, we can adopt a multi-step self-augmentation sampling strategy, to trade faithfulness for realism:

$$\tilde{\mathbf{x}}_{t_k}^{(k)} = (1 - t_k)\tilde{\mathbf{x}}_0^{(k)} + t_k\mathbf{x}_1 + t_k(1 - t_k)\sigma^2\epsilon_k \tag{3}$$

$$\tilde{\mathbf{x}}_0^{(k+1)} = \mathcal{E}_\theta\big(\tilde{\mathbf{x}}_{t_k}^{(k)}, t_k\big), \qquad k = 1, 2, \ldots \tag{4}$$

where $\tilde{\mathbf{x}}_0^{(k)}$ is the last estimation of CMR. $\tilde{\mathbf{x}}_{t_k}^{(k)}$ is the corresponding self-augmented sample. $t_k$ is a chosen time step $0 < t_k < 1$. Both sampling strategies are illustrated in Fig.2(b).

## 3. Experiments and Conclusions

We evaluated cHeartFlow on two types of auto-generated coarse sketches (edges/segmentations) as input, where the edges are detected by the Canny operator and the segmentations are generated using the Segment Anything model (SAM) (Kirillov et al., 2023). We used pix2pix (Isola et al., 2017) and iDDPM (Nichol and Dhariwal, 2021) with concatenated conditioning (Rombach et al., 2022) as the baseline methods. In Fig.1, we compare cHeartFlow's generation quality with the two baselines (upper panel) and show its robustness using previously unseen hand-drawn sketches (lower panel). In Table 1, we further report the generation quality of cHeartFlow using only one-step or two-step sampling, compared with pix2pix and iDDPM with much more iterations. In particular, the last two rows of Table 1 indicate an effective trade-off between faithfulness and realism, where a higher SSIM/PSNR score represents more faithful generation towards the original pixel distribution and a lower FID/LPIPS score denotes better perceptual alignment with the real images.

Table 1: Comparison of CMR generation qualities on M&Ms dataset with models trained on ACDC dataset. cHeartFlow outperforms the baseline methods with few number of function evaluations (NFE). Importantly, two-step generations (the last row) effectively trade faithfulness (higher SSIM / PSNR score) for realism (lower FID / LPIPS score).

| Method | NFE | FID↓ | SSIM↑ | PSNR↑ | LPIPS↓ |
|---|---|---|---|---|---|
| pix2pix | 1 | 238.0 | 0.5174 | 19.6232 | 0.2402 |
| iDDPM | 200 | 163.0 | 0.4820 | 12.3498 | 0.2317 |
| iDDPM | 1000 | 52.4 | 0.7934 | 25.0674 | 0.0558 |
| Ours (one-step) | 1 | 29.3 | **0.8521** | **28.2697** | 0.0701 |
| Ours (two-step) | 2 | **18.7** | 0.8331 | 27.5667 | **0.0555** |

To conclude, we introduced cHeartFlow, a novel consistency-model-based method for CMR generation trained by contrastive pairs between CMR and its auto-generated coarse sketches. We empirically demonstrated cHeartFlow's capacity to synthesize high-quality CMRs with flexibility and generalizability. This enables CMR image generation using coarse sketches with predetermined subject-specific anatomy, leading to controllable generation of patient cohorts, which potentially facilitates a variety of downstream learning tasks in medical imaging.

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
