# OpenReview forum: "cHeartFlow: Synthesizing cardiac MR images from sketches"
_MIDL.io/2024/Short_Papers — MIDL 2024 Short Papers_

### Official Review · Reviewer_KePk · 2024-04-23

**Confidence:** 4
**Final Rating:** 4

**Review:**

Key idea: The authors propose a novel generative framework to synthesize cardiac MR image conditioned on various different kinds of conditions such as segmentation masks, hand sketches, etc. The proposed approach is optimized in a contrastive learning manner such that the interpolations between encodings of condition and images are minimized. Using the proposed approach the images can be synthesized in even a single step.

1) Quality:
The authors provide good explanation of methods. Furthermore, comprehensive experiments are performed to evaluate the effectiveness of the method comparison with baseline models.
One limitation is the limited information regarding training of the networks in comparison to deep diffusion probabilistic models.

2) Clarity:
Overall, the paper is easy to follow. The goals are clearly stated and accordingly experiments are performed.  The only thing lacking is the limited information regarding training of the networks in comparison to deep diffusion probabilistic models.

3) Originality:
The proposed approach is novel in the medical imaging field.

4) Significance:
The generative model enables fast sampling (as few as 1 step compared to 200 in iDDPM). Furthermore, the model seems to work with different kinds of conditions, e.g., segmentation masks, hand drawn edges, etc.
Pro: + Fast sampling (1 or 2 step approach)
+ Reasonable alternative to DDPMs
+ Simple sketches as input allow more flexibility for synthesizing new samples

---

### Decision · Program_Chairs · 2024-04-26

Accept